



# Front-orography interactions during landfall of the New Year's Day Storm 1992

Clemens Spensberger[1,2] and Sebastian Schemm[3,2]

[1]Geophysical Institute, University of Bergen and Bjerknes Centre for Climate Research, Bergen, Norway
[2]Institute for Atmospheric and Climate Science, ETH Zurich, Zurich, Switzerland
[3]Laboratoire de Météorologie Dynamique/IPSL, École Normale Supérieure/CNRS/UPMC, Paris

**Correspondence:** C. Spensberger (clemens.spensberger@uib.no)

**Abstract.** Although following a common synoptic evolution for this region, the New Year's Day Storm 1992 was associated with some of the strongest winds observed along the Norwegian West Cost. The narrow wind band along its bent-back front became famous as the "poisonous tail", and paved the way towards today's sting jet terminology. This article re-examines the storm's landfall with a particular focus on the interaction with the orography.

Sensitivity analyses based on WRF simulations demonstrate that the formation and the evolution of the warm-air seclusion and its "poisonous tail" are largely independent from orography. In contrast, the warm sector of the storm is undergoing considerable orographically induced modifications. Both warm and cold fronts are eroded rapidly, and the warm sector is lifted over the orography, thereby accelerating the occlusion process. The insensitivity of the warm-air seclusion to the orographic modifications of the warm sector raises the question to which extent these entities are still interacting after the onset of the

occlusion process.

Further, we observe ubiquitous and large-amplitude internal gravity waves (IGWs) during the landfall of the warm and cold fronts, exceeding in amplitude the cross-frontal circulation. As the spatial scales of the IGW pattern and of the fronts are comparable, we speculate that wave-front interactions might have contributed to the rapid erosion of the cross-frontal temperature gradient over the orography. Further, IGWs might also provide a plausible cause for the observed near-instantaneous flow

deflection around orography at 500 hPa, well above crest height.

## 1   Introduction

The New Year's Day Storm 1992 (norw.: *Nyttårsorkanen*) was one of the most vigorous winter storms in the history of recorded storms over Norway. Its wind gusts exceeded the upper limits of measurement stations along the Norwegian West Coast.

Through extrapolation, the Norwegian national weather service (MET Norway) estimated wind gusts exceeding $60\,\mathrm{m\,s^{-1}}$ and 10-min mean wind speeds exceeding $45\,\mathrm{m\,s^{-1}}$ at Svinøy lighthouse during the landfall (locations and orography shown in Fig.



1; Aune and Harstveit, 1992). Along the coastline from the Svinøy Lighthouse to the vicinity of Trondheim, the wind speeds are estimated to have return periods of more than 200 years (Aune and Harstveit, 1992).

Consequently, the storm attracted the attention of the scientific community. In particular the formation of the extreme wind
speeds along the bent-back front and the formation of a warm-air seclusion received attention (Grønås, 1995; Browning, 2004; Clark et al., 2005). With his study, Grønås (1995) introduced the "poisonous tail" as a widely-used metaphor to the scientific literature. The term "poisonous tail" was already used by to the Norwegian forecasting community before the work of Grønås (1995), who learned about it while working as a forecaster at MET Norway in the 1960s. Later, Browning (2004) and Clark et al. (2005) turned the "poisonous tail" into the "sting at the end of the tail", which has become a term popular even in the media
(Schultz and Browning, 2017). In his analysis of the New Year's Day Storm, Grønås (1995) found that moist diabatic processes played a crucial role in the formation and evolution of the warm-air seclusion. We complement his analysis, investigating the role of front-orography interactions in the evolution of the storm. Specifically, we investigate the role of the landfall on the Scandinavian coastal range (Scandes) in the formation and evolution of the warm-air seclusion.

Previous case studies on front-orography interactions often focus only on cold fronts (e.g. Braun et al., 1997; Kljun et al.,
2001; Neiman et al., 2004), although a limited number also consider bent-back fronts (e.g. Steenburgh and Mass, 1996) or warm fronts (e.g. Doyle and Bond, 2001). Together, these case studies document numerous processes that can be at play during front-orography interactions, for example front dissipation due to orographic flow deformation (Braun et al., 1997), the formation of barrier jets (Braun et al., 1999; Yang et al., 2017), front retardation due to blocking (Doyle and Bond, 2001; Neiman et al., 2004), as well as foehn and cyclogenesis in the lee (Kljun et al., 2001).

Due to the complexity of the front-orography interplay, only a limited number of conceptual models exist. One of the earliest efforts is presented by Bjerknes and Solberg (1921), who summarised rain observations over Scandinavia in a conceptual model of a warm front moving over a ridge. Their conceptual model includes a pocket of trapped cold air upstream of the ridge and a warm front modified by the vertical motion up- and downstream of the ridge. Davies (1984) and Egger (1992) apply the idea of orographically blocked flow to a non-linear analytical model of cold fronts approaching a triangular ridge. In their model,
the approaching cold front is entirely blocked in a certain parameter regime, for example for strong stratification. In contrast, if the flow is less stratified, the approaching front might be linearly advected by flow over the ridge (Blumen and Gross, 1987; Blumen, 1992). Here the flow over the ridge causes a weakening of the front upstream, and a strengthening downstream. The antipodal conceptual models of Davies (1984) and Blumen and Gross (1987) provide the framework within which we will interpret our results.

As is shown in this study, the synoptic evolution of the New Year's Day Storm during landfall, including the formation of the warm-air seclusion, closely follows the conceptual model in Fig. 7 of Bjerknes and Solberg (1922). Two aspects make this similarity noteworthy. First, it indicates that the storm followed a common synoptic evolution for this region. This seems surprising for such an extreme event. If the evolution followed a common pattern, why did this particular storm produce low-level winds with return periods of 200 years or more? While we do not claim to provide the ultimate answer, our analyses allow
an hypothesis on what separates the New Year Day's Storm from more average intense winter storms in the region. Second, the conceptual model in Fig. 7 of Bjerknes and Solberg (1922) is only a special case of their conceptual model of secondary



cyclogenesis at the occlusion point (their Fig. 6). Hence, as Bjerknes and Solberg (1922) were well aware, the formation of a warm-air seclusion can occur also without any orographic influence. This raises the question to which extent the warm-air seclusion of the New Year's Day Storm formed due to the Scandes. Here, a sensitivity analysis allows us to draw a definitive

conclusion.

## 2 Data and methods

We base our analysis on 3-hourly data from the Norwegian Reanalysis Archive (NORA10; Reistad et al., 2011) as well as on sensitivity experiments using the Weather Research and Forecasting model (WRF; Skamarock et al., 2008). The horizontal resolution of the NORA10 data set varies between 10 and 11 km, with the NORA10 orography and domain boundaries indi-

cated in Fig. 1. In the vertical, the data is interpolated from 40 hybrid model levels to 25 pressure levels, covering the lower troposphere (1000–700 hPa) with a uniform resolution of 25 hPa, and a resolution of 50 hPa above. In the course of our analyses we noted that for this explosively developing case and the time period we consider in this study, the NORA10 hindcast appears to lag behind the actual development as seen on satellite imagery by about 1-1.5 hours. We therefore refer to the respective time scales as $UTC_{No}$ for the hindcast and $UTC_{Sat}$ for the satellite imagery where ambiguous.

### 2.1 WRF control and sensitivity simulations

The WRF model setup parallels that of the NORA10 hindcasts. We use a constant horizontal resolution 10 km, but the WRF standard configuration with 60 hybrid model levels in the vertical. The output is interpolated to the same 25 pressure levels as the NORA10 data. The domain horizontally covers an area of $6000 \times 4000$ km, including most of the North Atlantic and of Scandinavia. The domain fully covers the region shown in Fig. 1. The Norwegian West Coast is located centrally in the eastern

half of the model domain to minimise the impact of the model boundaries.

The initial state and boundary conditions are derived from 6-hourly ECMWF operational analyses. As the operational analyses for the study period do not include sea-surface temperature, we supplement with those from ERA-Interim (Dee et al., 2011). We initialise the model on 31 December 1991 06 UTC, roughly 21, 24, and 27 hours before the landfall of the warm, cold, and bent-back fronts, respectively. We chose this initialisation time as a trade off between having sufficient model spin-up

time and the accuracy of the simulation. We tested initialisation dates between 30 December 1991 12 UTC and 31 December 1991 UTC and find that the earlier the initialisation date, the weaker becomes the simulated New Year's Day Storm.

We use WRF version 3.8.1 in a configuration with YSU boundary layer physics, the revised MM5 scheme (based on a Monin Obukhov surface layer with Carlson-Boland viscous sublayer), and the WRF three-class single moment cloud microphysics scheme. The parameterisation of cumulus convection is enabled, using the Kain-Fritsch scheme updated every 5 minutes.

Radiation is calculated using the RRTM for long wave lengths, and the Dudhia scheme for short wave lengths, both updated every 10 minutes. The land surface is parameterised using a thermal diffusion model with five layers.

To test the sensitivity of the evolution of the New Year's Day Storm to the Scandinavian orography, we compare the control simulation with full orography to (a) a simulation in which the orography is removed and the land surface replaced by ocean,





and (b) a simulation in which we double the height of the Scandes. In the Ocean simulation, sea-surface temperatures over the
Scandinavian peninsula are defined by iteratively minimising the local Laplacian given the observed surface temperatures in
the surrounding seas (277-280 K in the Atlantic and North Sea ; 275-278 K in the Baltic; 273-275 K in the White Sea).

## 2.2 Front detection

For our analysis of the landfall, we require a front detection scheme to track and visualise the evolution of the fronts close to
orography in the comparatively high-resolution NORA10 hindcasts and WRF simulations. There are different approaches for
the automated detection of fronts, but all of them struggle to detect fronts in the vicinity of orography. Many of the difficulties
arise from the need of higher derivatives to pinpoint the exact location of a front line (e.g. in Hewson, 1998; Jenkner et al.,
2010; Berry et al., 2011). We therefore follow the approach of Spensberger and Sprenger (2018) and detect frontal volumes
instead of front lines. Frontal volumes are defined as coherent volumes where the local thermodynamic gradient exceeds a
given threshold

$$|\nabla \tau| > K \,, \tag{1}$$

and that exceed a given minimum volume. With this approach, we require only a first derivative of a thermodynamic field.

Spensberger and Sprenger (2018) developed this approach to detect fronts in the ERA-Interim data set. In our comparatively
high-resolution data sets, mesoscale processes can lead to locally strong thermodynamic gradients independent from a synoptic-
scale frontal system. We are however only interested in local thermodynamic gradients associated with a synoptic-scale front,
and therefore require a minimum thermodynamic gradient both at the native resolution of the input data and in a smoothed
version of the thermodynamic field. With this extension, we are able to identify only those mesoscale gradients that belong to
a synoptic scale system, and define only those as fronts.

For the thermodynamic field $\tau$, we use equivalent potential temperature $\theta_e$ to include both temperature and moisture gradients
in the front definition [see Schemm et al. (2017) and Thomas and Schultz (2019) for a comprehensive discussion on the merits
and drawbacks of $\theta_e$ for objective front detection]. We detect frontal volumes between 700 and 950 hPa with a minimum volume
of $7500 \, \text{km}^2 \cdot 250 \, \text{hPa}$. Furthermore, we use a local $|\nabla \theta_e|$-threshold of $6.0 \, \text{K} \, (100 \, \text{km})^{-1}$ and a threshold for the smoothed field
of $4.5 \, \text{K} \, (100 \, \text{km})^{-1}$. The conclusions of this study are independent from the exact values used for these thresholds.

To arrive at the lower-resolution data set, we smooth the $\theta_e$ field by 30 passes of a three-point filter $\tau_i^* = \tau_i + (\frac{1}{4}\tau_{i-1} - \frac{1}{2}\tau_i + \frac{1}{4}\tau_{i+1})$ in both horizontal dimensions. Here, $i$ is the grid point index and $*$ denotes the new value after one pass of the
filter. With 30 passes, this filter largely suppresses waves shorter than 10 grid points (approx. 110 km), while waves longer than
approximately 20 grid points retain more than 50% of their amplitude.





## 3 Synoptic evolution during landfall

### 3.1 Explosive deepening over the central North Atlantic

We start following the synoptic evolution at 00 UTC$_{No}$ on 1 January 1992, corresponding to 18 hours lead time of the WRF

simulations (Fig. 2). At this point in time, the cyclone underwent an explosive deepening of approximately 40 hPa during the preceding 24 hours (Grønås, 1995). Its structure is that of a mature cyclone following the Shapiro and Keyser (1990) conceptual model (Fig. 2a, b). The warm sector is bounded on the northern side by a well-developed bent-back front, and to the west by several cold front segments. On the warm front side, however, the $\theta_e$ gradient is below the detection threshold.

At this point in time, the strongest winds occur along the cold front (Fig. 3a, b), in agreement with, for example, the low-

level jet described in Lackmann (2002). This low-level jet ahead of the cold front transports warm moist air towards the cyclone core, providing the inflow to a warm conveyor belt, as indicated by the broad area of stratiform ascent along the bent-back and warm fronts (vertical wind in Fig. 4a, b; cloud observations in Fig. 5). Although the absolute wind speed is smaller along the developing bent-back front than along the cold front jet, stronger ageostrophic winds indicate considerable flow imbalances along the developing bent-back front (exceeding $20\,\mathrm{m\,s^{-1}}$ in the NORA10 hindcasts and $25\,\mathrm{m\,s^{-1}}$ in the WRF

control simulation; Fig. 3a, b).

### 3.2 Landfall of the warm sector

At 03 UTC on 1 January 1992, the leading edge of the warm sector arrives at the Scandes (Fig. 2c, d). At 850 hPa, the warm front does not fulfil our detection criteria except for a small region close to Svinøy on the northwestern cape of the West Coast, but at 925 hPa the temperature gradient exceeds the detection threshold in a line connecting Bergen with the cyclone's bent

back front (not shown). The temperature gradient tightens as the front approaches the coast line, consistent with increased $\theta_e$ frontogenesis at 850 hPa driven by flow deformation (not shown). This flow deformation is consistent with an increase in ascent long the warm front as it approaches the Scandes (Fig. 4c, d).

The intensification of the fronts approaching the Scandes indicates some orographic blocking of the warm air mass that acts frontogenetically. The orographic flow distortion is associated with increasing ageostrophic wind components where the

Scandes intersect with the 850 hPa surface (Fig. 3c, d). In contrast, at 700 hPa, parts of the warm sector already moved over the Scandes (not shown), which reach up to a crest height of about 850-800 hPa. Both the 700 hPa $\theta_e$ distribution and the corresponding cloud cover on the Meteosat 4 satellite imagery show hardly any orographic distortion (Fig. 5a), confirming that only levels below 850 hPa are affected by some degree of orographic blocking.

Between 03 and 06 UTC, the cold front crosses the North Sea and is at 06 UTC about to make landfall on the Norwegian

West Coast (Fig. 2e, f). With the approaching cold front, the warm sector narrows to a thin filament along the coast line north of Bergen. To the south of Bergen, the Scandes are lower and the warm sector continues to propagate eastward relatively unaffected by the orography. This north-south difference in propagation speed further indicates that the warm front is partly blocked to the north of Bergen. At 850 hPa and below, the core of the cyclone is already largely cut off from the warm sector (Fig. 2e, f), while at 700 hPa core and warm sector remain connected (not shown).



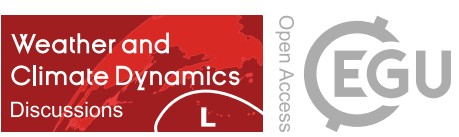

Along the coast line between Svinøy and Trondheim (Fig. 1), both the NORA10 hindcasts and the WRF control simulation exhibit an area of increased wind speeds (Fig. 3e, f). In WRF this area is a homogeneous tongue of high wind speeds associated with descent at 700 hPa (Fig. 4f). In contrast, the NORA10 hindcasts show pronounced small-scale variations in the horizontal wind speed collocated with varying up- and downdrafts (Figs. 3e, 4e).

During the landfall of the cold front, between 0430 UTC$_{Sat}$ to 0600 UTC$_{Sat}$, satellite imagery shows the development of a
cloud-free area on the eastern side of the Scandes (Fig. 5d). This cloud-free area indicates descending air masses and cloud evaporation. There is however some indication that this descent does not reach the lowest troposphere. First, the southwesterlies in the warm sector at 850 hPa are not evident in the wind field in the lee of the Scandes (Fig. 2e, f). Second, the 850 hPa temperature in this region remains unchanged between 03 UTC and 06 UTC$_{No}$ (Fig. 2c-f), suggesting that the lowest levels in the lee of the Scandes are at 06 UTC still covered by the incipient cold air mass.

Although some of the cold air moved over the Scandes after 09 UTC (Fig. 2g-j), the temperature gradient in the lee has become too diffuse to qualify as a cold front following our definition. Further, the previously clear signal in the vertical wind associated with the warm front (Figs. 4c, d) disappears over the Scandes. Over and in the lee of the Scandes, the vertical wind pattern is now dominated by wave structures indicating considerable activity of orographically triggered gravity waves (IGWs; Figs. 4c-h). These waves dominate over the organised vertical wind structure associated with the frontal circulation over the
Atlantic and North Sea. The larger scale structure of the horizontal wind in the warm sector is re-emerging in the lee of the Scandes (compare Figs. 3c, d and 3g, h), although the peak wind speed is considerably reduced and features a superposed a wavy pattern consistent with IGWs. In contrast, over southern Sweden and Denmark, the cold front could propagate eastward without encountering any orography, leaving the wind structure largely intact (Figs. 3g, h).

### 3.3 Evolution of the warm-air seclusion

So far we focused the synoptic discussion on the landfall of the cold and warm front, and orographic impacts on the warm sector. In the following, we shift focus towards the remaining front structure, the bent-back front, and its evolution in tandem with the warm-air seclusion.

At 09 UTC$_{No}$, the warm-air seclusion is fully cut off from the warm sector at 850 hPa (Fig. 2g, h). The warm sector is located entirely on the eastern side of the Scandes, and covers the southern part of Scandinavia (Fig 2g, h). At 700 hPa, however, the
warm-air seclusion is still largely connected with the warm sector, although the warm-air seclusion is now associated with a separate $\theta_e$ maximum close to the bent-back front (Fig. 6a, b).

With the formation of the warm-air seclusion, the cloud cover in the cyclone core starts to change. At 0300 UTC$_{Sat}$, the cyclone core is covered by low-level clouds, and partly even cloud-free. From 0430 UTC$_{Sat}$ onward, the cyclone core is increasingly covered by spots of high-top convective clouds (Fig. 5c-e). The temporal correlation with the formation of the
warm-air seclusion indicates that the onset of the convective activity in the cyclone core is dynamically linked with the cut-off of the seclusion from its warm sector. At 0730 UTC$_{Sat}$ most of the warm-air seclusion is covered by patchy high clouds (Fig. 5e).



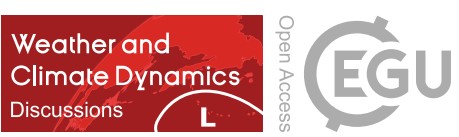

From 00 UTC to 09 UTC, the bent-back front continuously changes orientation. At 00 UTC the tip of the bent-back front points to the southwest, at 06 UTC to the south, and at 09 UTC to the southeast. The change of orientation is particularly
pronounced between 06 UTC and 09 UTC, while the tip of the bent-back front is rapidly approaching the coast line. Between 09 UTC and 12 UTC, the bent-back front makes landfall and its orientation stops changing. Thus, the low-level jet associated with the bent-back front impinges on the coast line almost perpendicularly (Fig. 2d, e).

The following hours are characterised by a rapid decay of the warm-core seclusion (not shown). The weakening cyclone core crosses the Scandes between 15 and 18 UTC at around 65°N, where the Scandes are lower than further south. Together
with the cyclone core, parts of the warm-air seclusion cross the Scandes, but in this process the temperature gradient along the bent-back front weakens, and no longer qualifies as a front following our definition.

### 3.4   Assessing the WRF control simulation

Overall, WRF reproduces the synoptic evolution of the New Year's Day Storm well compared to the NORA10 reanalyses and the satellite imagery (Figs. 2-6). As indicated in the discussion of the synoptic evolution, there are however more or less subtle
differences between NORA10 and WRF. Conceptually, the most prominent difference concerns the dynamics of the flow in the lee of the Scandes, between Svinøy and Trondheim at 06 and 09 UTC (Figs. 3e-h; 4e-h). Here, WRF simulates a coherent region of descent which might be indicative of a foehn event, while the NORA10 hindcasts exhibit patchy up- and downdrafts indicative of convective activity. This and other differences between NORA10 and WRF are however inconsequential for the following discussion of orographic effects.

## 4   Orographic impacts

### 4.1   Little impact on the formation of the warm-air seclusion and its poisonous tail

In the synoptic evolution of the New Year's Day Storm in NORA10 and WRF, we observe clear orographic influence on the storm. In particular, we found a separation of the cyclone core from its warm sector below 850 hPa, a synoptic evolution similar to the one described in Bjerknes and Solberg (1922). However, the extent to which the formation of the warm-air seclusion
was forced or accelerated by the orography remains unclear. Levels above 850 hPa did not experience an orographically forced cut-off of a pocket of warm air from the warm sector but still see a warm-air seclusion. To assess the impact of orography on the storm, we performed sensitivity analyses in which we either replaced the Scandinavian peninsula by ocean ("Ocean" simulation), or doubled the height of the Scandes ("Double" simulation).

These changes to the orography have surprisingly little impact on the evolution of the storm (Fig. 7). In particular, these
simulations demonstrate that the warm-air seclusion would have formed even in the absence of orography (Fig. 7a). Without orography, the warm sector would have deformed to a long arched filament on 850 hPa (Fig. 7a), indicating an ongoing occlusion process independent from orography. The Scandes however accelerated this deforming and separation in an orographic occlusion process.





In tandem with the largely unaffected warm-air seclusion, the bent-back front shows hardly any orographic impact prior to its
own landfall around 09 UTC (Fig. 7). In the Ocean simulation, the detected bent-back front extends further around the warm-air seclusion, indicating a more locally confined $\theta_e$ gradient on the northern side of the seclusion without orographic influence.
The structure of the wind field around the poisonous tail however are largely unaffected by the orography (not shown), and in
particular unaffected by the landfall and decay of the storm's warm sector.

This insensitivity of the wind field might indicate that the extreme winds in the storm's poisonous tail are part of a sting jet,
dynamically arising from downward transport of momentum (Schultz and Browning, 2017; Clark and Gray, 2018). However,
in neither the NORA10 hindcasts nor our WRF simulations, there is a region of coherent descent that would indicate a sting
jet (Fig. 4c-h). We therefore interpret the poisonous tail as predominantly a cold conveyor belt jet (Clark and Gray, 2018). In
synthesis with the results of Grønås (1995), we suggest that the dynamics of the poisonous tail are more determined by local
moist diabatic effects than somewhat more remote orographic flow distortions.

In summary, the warm-air seclusion and its poisonous tail exhibit only weak sensitivity to the partial cut-off and rapid decay
of the warm sector. This weak sensitivity suggests that the cyclone core and its warm sector hardly interact at this stage in the
development. The fronts fronts framing the warm sector decay quickly when moving over the Scandes and the warm conveyor
belt inflow is at least partially interrupted. Yet, the warm-air seclusion evolves largely unaffected. Consequently, it might be
most appropriate to regard the seclusion and the warm sector as two dynamically independent entities as soon as the occlusion
process started. The evolution of these entities would then be in sync primarily because of a joint history rather than because
of a persistent dynamical linkage.

The observation of Bjerknes and Solberg (1922) that secondary cyclogenesis frequently occurs at the occlusion point cor-roborates our interpretation. Cyclogenetic processes in the warm section frequently lead to the formation of a secondary core
rather than intensifying the incipient cyclone. Secondary cyclogenesis thus provides further indication that the dynamical ties
between the warm sector and the incipient cyclone core tend to weaken considerably with the onset of the occlusion process.

## 4.2 Northward displacement of the warm-air seclusion

The main impact of the Scandes on the warm-air seclusion is a slight displacement to the north in the presence of orography
that goes along with higher pressure on the upstream side of the Scandes. These orographically induced differences in the
pressure (and hence mass) distribution are consistent with the orographically impacted mass transports in Figure 8. Figure 8
shows the mass transport $\sigma$ at 850 hPa perpendicular to the geopotential isolines at this level. Here,

$$\sigma = -\rho \boldsymbol{v} \cdot \frac{\nabla \phi}{|\nabla \phi|} \quad , \tag{2}$$

where $\rho$ is density, $\boldsymbol{v}$ the horizontal wind vector and $\phi$ geopotential.

Before landfall, mass transport towards the lower geopotential occurs primarily along the cold front as well as a convergence
line in the cold sector (Fig. 8a, b). Further, the mass transport exhibits a pronounced dipole structure around the cyclone core,
which we expect mainly reflects the movement of the low pressure centre.





Even before the landfall of the fronts, the Scandes induce some mass transport towards lower pressure along the Norwegian west coast, peaking close to Svinøy (Fig. 8b). The mass transport peaks around 03 UTC (Fig. 8d), the time of the landfall of the warm front and the associated start of the orographic occlusion process. It is still a prominent feature of the mass transport around 06 UTC (Fig. 8f), when the orographic occlusion process completed the separation of the cyclone core from its warm

sector. It is plausible that this orographically induced mass transport caused both the displacement of the cyclone core to the north, as well as the higher pressures along the Norwegian west coast.

Interestingly, this orographic mass transport is not restricted to the height of the orography. Although it is most pronounced below crest level, it remains visible throughout the troposphere, as for example at 500 hPa, shown in Figure 9. At this level, the orographic impact is evident in a dipole pattern centred over the Scandes in southern Norway that is consistent with a ridge

evolving over the orography.

While the appearance of this orographically induced mass transport in the middle troposphere is hardly surprising, it might be instructive to ask what process causes its appearance. The 500 hPa mass transport evolves in tandem with the one at 850 hPa without discernible delay (shown for 03 UTC, compare Figs. 8d, 9b). Whatever does communicate the orographic impact to 500 hPa must hence do so rapidly. In the light of the ubiquitous and high-amplitude IGW signatures in the vertical wind over

and in the lee of the Scandes (Fig. 4), it seems plausible that IGWs are responsible for the changing mid-tropospheric flow pattern.

### 4.3 Considerable impact on the warm and cold fronts

IGWs might also have impacted the evolution of the warm sector, and in particular the cold front. The cross-front temperature contrast is considerably smoother in the lee of the Scandes (Fig. 2), suggesting that the cold front largely decayed while passing

over the Scandes. Further, over and in the lee of the mountains the vertical wind does no longer show any signs of the frontal circulation (Fig. 4e-h). As the wave length of the IGWs over the Scandes is comparable to the cross-frontal length scale of the approaching cold front, interactions between the frontal circulation and IGWs seem likely. This interaction would need to be non-linear in order for IGWs to be able to alter the background state on which they propagate (e.g., review on the sources and effects of IGWs in Plougonven and Zhang, 2014).

In order to investigate further, we follow the evolution of the warm and cold fronts in cross sections through the warm sector in the Ocean and Double simulations (Figs. 10, 11). Without orography, the cold sector catches up with the warm sector around 05 UTC, and an occlusion process begins (Fig. 10b). In this simulation ageostrophic winds exceeding $15\,\mathrm{m\,s^{-1}}$ occur solely in the boundary layer and along the eastward end of a wind maximum in the upper troposphere, around 400 km along the section in Fig. 10b.

In the Double simulation, the equivalent potential temperature and wind structure differs considerably from that in the Ocean simulation (Fig. 11). Most prominently, upstream of the Scandes a cold anomaly exceeding 8 K is evident both at 03 and 05 UTC. This cold anomaly indicates the orographically retarded propagation of the warm front at 03 UTC and a leftover pocket of incipient cold air trapped below 900 hPa at 05 UTC (Fig. 11a, b), similar to the one included the conceptual model of Bjerknes and Solberg (1921). Both at 03 UTC and at 05 UTC, low-level ageostrophic winds upstream of the Scandes exceed $35\,\mathrm{m\,s^{-1}}$.





Throughout the shown evolution, wavy patterns in the ageostrophic wind component indicate IGW activity over the leeward slopes of the Scandes (Fig. 11). Consistent with our interpretation as IGWs, these wavy patterns are even more pronounced in the stratosphere and there exceed amplitudes of 20 K in $\theta_e$.

In the same region, isentropes are pulled down to follow the orographic slope over varying fractions of the troposphere (Fig. 11a-d). These simulated downdrafts are consistent with the appearing cloud-free area observed in the satellite imagery after the

passing of the cold front (Fig. 5). Despite these downslope winds, neither the warm nor the cold sector affect the lowest level temperatures in the lee of the Scandes. This is particularly evident at 07 UTC, when the warm sector is located in the region between 800 km and 1000 km along the section in the Ocean simulation (Fig. 10c). In the same region, there is a pronounced cold anomaly below crest height in Figure 11c, showing that the orographically lifted warm sector hardly descends in the lee of the Scandes. This result confirms our previous interpretation of the orographic impact on the warm sector as an orographic

occlusion.

A trajectory analysis based on the NORA10 hindcasts provides further support. Air parcels released in the North Sea upstream of the Norwegian west coast at 925 hPa in (a) the incipient cold air mass, (b) the warm sector and (c) the cold sector all are first lifted above the Scandes and then either level off or continue their ascent. Only few trajectories released in the cold sector descend the lee slopes of the Scandes. The warm sector is thus lifted off the surface in the lee of the Scandes.

Based on the demonstrated orographic impact on the warm sector, we can finally form a hypothesis on why this particular storm was associated with extreme winds despite following a relatively typical pattern (Bjerknes and Solberg, 1922). Comparing our simulations, we note that the landfall of the warm sector in the Double and Control simulations coincides almost perfectly with the onset of the occlusion process in the Ocean simulation. As the occlusion process is only somewhat accelerated in the Double and Control simulations, the forming warm-core seclusion could evolve largely unaffected by orography.

Further, the landfall of the bent-back front a few hours after the begin of the orographic occlusion coincides very well with the most intense stage of the cyclone, which also for the New Year's Day Storm is observed slightly after the begin of the occlusion process (more generally documented in Schultz and Vaughan, 2011). From these findings, we hypothesise that it was mainly the timing of the landfall within the life cycle of the New Year's Day Storm that made this case such an extreme event.

## 5 Summary and conclusions

In summary, we followed the synoptic evolution of the New Year's Day Storm from its mature stage and through the landfall of, in sequence, the warm, cold and bent-back front. Perhaps surprisingly, we find that the formation and evolution of the warm-air seclusion, the feature that Grønås (1995) made responsible for the storms devastating effects, is largely independent from the Scandinavian coastal range (Scandes). This insensitivity is largely due to the cyclone's inherent occlusion process. The Scandes did hence not induce, but only accelerate the occlusion process without considerably affecting the formation and

evolution of the warm-air seclusion. Further, we find that the extreme wind speeds along the bent-back front of the storm, the poisonous tail, form as a cold conveyor belt jet and occur irrespective of orography.

With that finding, the New Year's Day Storm becomes an ideal natural experiment to clarify the dynamical relation between the cyclone core (here, the warm-air seclusion) and its warm sector once the occlusion process has started. While the Scandes clearly affect the warm sector by retarding and eroding the fronts that move over the orography, the evolution of the warm-air

seclusion is only slightly displaced to the north, but otherwise unaffected. This insensitivity of the core to the evolution of the warm sector suggests that the cyclone core at this stage of the life cycle has become a largely independent dynamical entity, that only co-evolves rather than interacts, with its warm sector. The observed tendency for secondary cyclogenesis at the occlusion point indicates that this finding for the New Year's Day Storm might apply more generally.

Based on these findings, we speculate that the timing of the landfall within the life cycle of the New Year's Day Storm

was crucial to make it an extreme event. The landfall of the warm sector coincided well with the natural onset of the storm's occlusion process, thereby limiting the effect of the landfall on the formation and evolution of the warm-air seclusion. Further, the landfall of the bent-back front coincided well with the peak intensity of the storm, a few hours after the beginning of the occlusion.

In this study, we could not clarify why the evolution of the New Year's Day Storm constitutes a common pattern for the

region. For this pattern to appear, storms which approach Norway would generally need to be at a similar stage of their life cycle. A climatological quantification of typical genesis regions for these storms could clarify this and put this case study in a climatological context.

Finally, we highlighted two examples where internal gravity waves (IGWs) might have played a role in the synoptic evolution of the storm. First, they might be responsible for the orographically induced mass transport in the mid-troposphere that

contributed to the northward displacement of the warm-air seclusion. Second, they might have contributed to the erosion of the cold front while it passed over the Scandes. To underpin these more tentative results, and generalise them beyond the case study presented here, we would require a more systematic assessment of the role of IGWs for front-orography interactions.

*Author contributions.* Both authors contributed to the analysis and writing of the manuscript.

*Competing interests.* We declare no competing interests.

*Acknowledgements.* We thank Thomas Spengler, Lukas Papritz, and Heini Wernli for interesting discussions, the Norwegian Meteorological Institute for providing the NORA10 reanalysis, the University Cooperation for Atmospheric Research (UCAR) for developing and providing the WRF model, the European Centre for Medium-Range Forecasts (ECMWF) for providing the boundary conditions used for the WRF simulations, and EUMETSAT for the Meteosat 4 imagery.



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



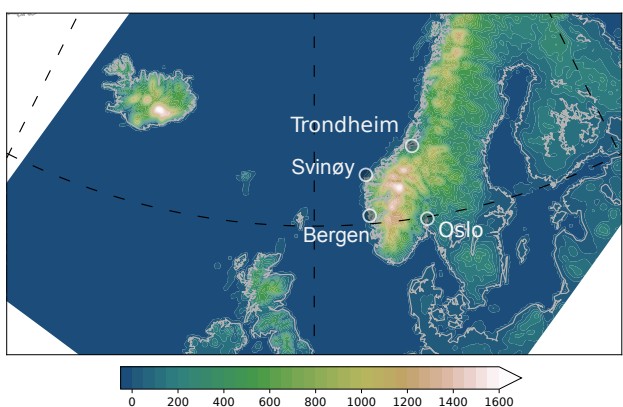

**Figure 1.** Orographic height map [m] from the NORA10 data set. The white circles show the location of some major cities and the Svinøy Lighthouse mentioned in the text.



**Figure 2.** Frontal evolution for the New Year's Day Storm at 850 hPa. Equivalent potential temperature shading [K] and geopotential height contours with an interval of 50 m. Detected frontal zones are marked by dark shading. The panels show the development from (top row) 00 UTC through (bottom row) 12 UTC on 1 January 1992 in 3-hour intervals. The left column is based on the NORA10 hindcasts, the right on a WRF simulation.



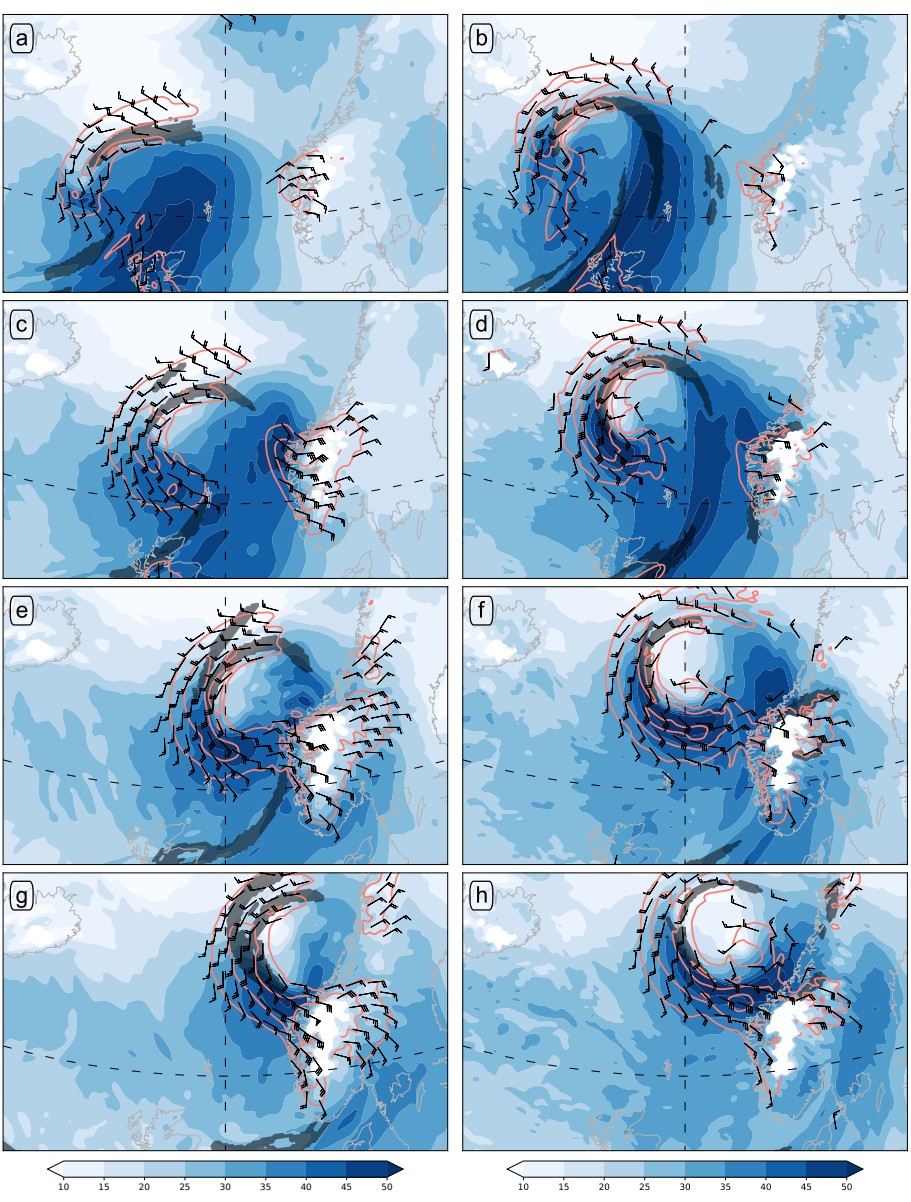

**Figure 3.** Full wind speed $[\mathrm{m\,s^{-1}}]$ at 850 hPa on (a,b) 00 UTC, (c,d) 03 UTC, (e,f) 06 UTC and (g,h) 09 UTC of 1 January 1992. Barbs and pale red contours show ageostrophic wind with contours at 15, 25, and $35\,\mathrm{m\,s^{-1}}$. Dark shared regions show location of front volumes at 850 hPa as in Fig. 2. The left column is based on the NORA10 hindcasts, the right on a WRF simulation.



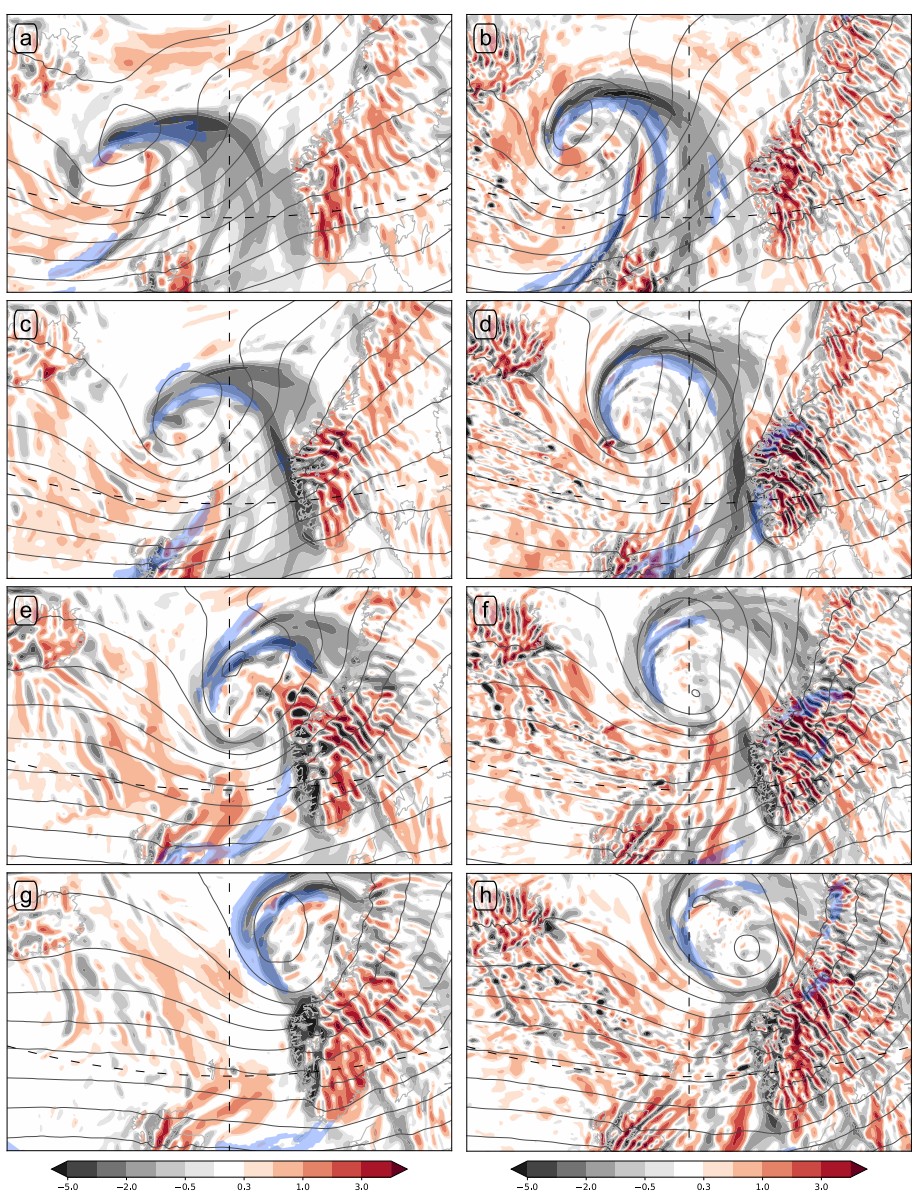

**Figure 4.** Pressure vertical velocity [Pa s$^{-1}$] at 700 hPa for (a,b) 00 UTC, (c,d) 03 UTC, (e,f) 06 UTC, and (g,h) 09 UTC. Dark grey contours show geopotential at 700 hPa, with a contour interval of 50 m, respectively. Blue shaded regions indicate the location of frontal zones at 850 hPa for both columns. The left column is based on the NORA10 hindcasts, the right on a WRF simulation.



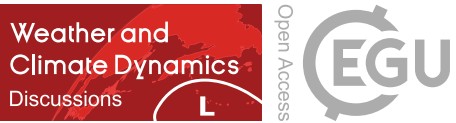

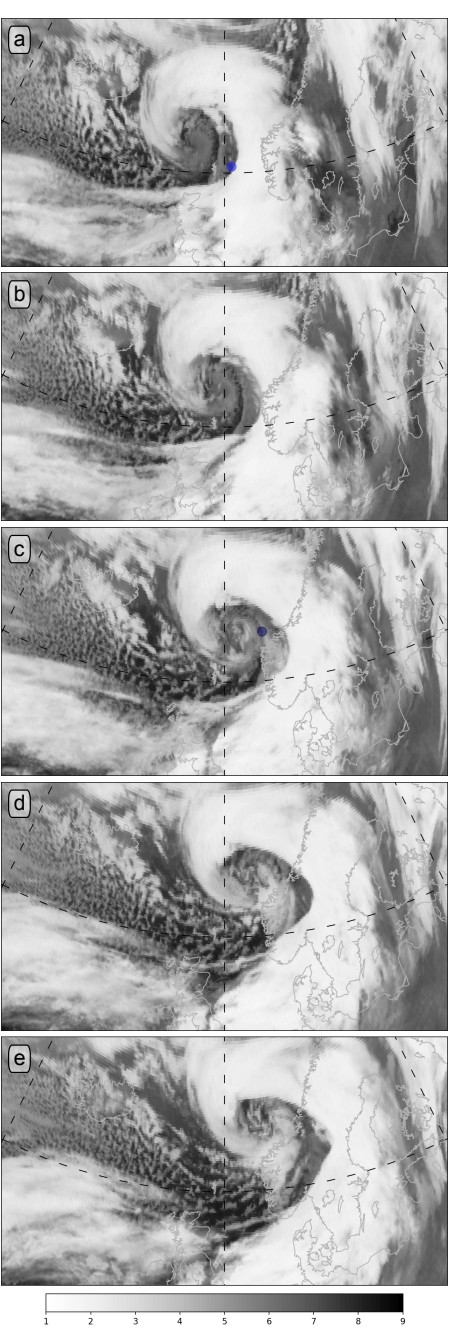

**Figure 5.** Remapped infrared imagery (10.8μm wave length, in $\mathrm{W\,m^{-2}\,sr^{-1}}$) from Meteosat 4 covering the landfall of the New Year's Day Storm from (a) 0130 $\mathrm{UTC_{Sat}}$ through (e) 0730 $\mathrm{UTC_{Sat}}$ on 1 January 1992 in 1.5-hour intervals. These times correspond approximately to the period 03 $\mathrm{UTC_{No}}$–09 $\mathrm{UTC_{No}}$. Imagery © EUMETSAT 2016.





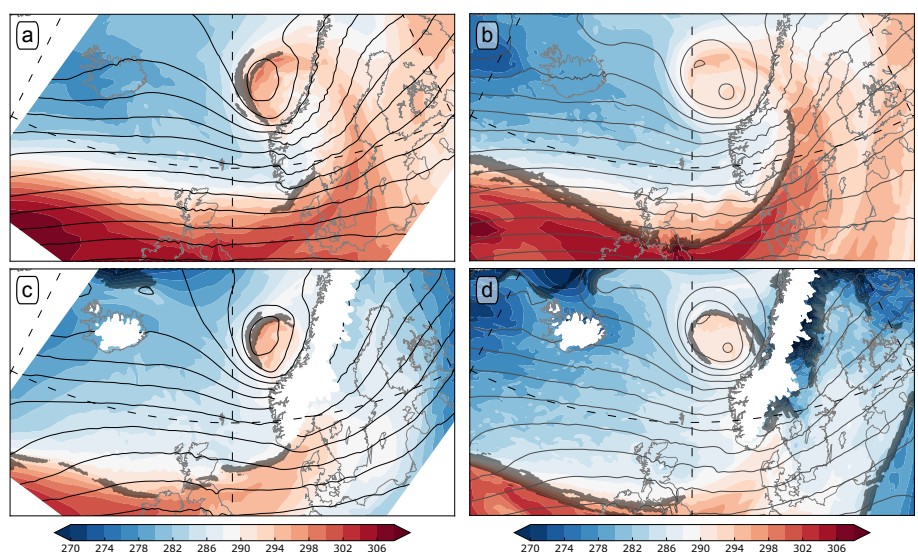

**Figure 6.** As Fig. 2g,h, but for (a,b) 700 hPa and (c,d) 925 hPa.

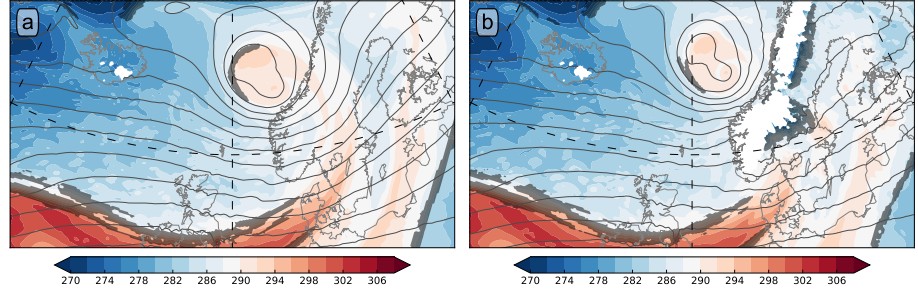

**Figure 7.** As Fig. 2g,h, but for (a) the Ocean simulation, and (b) the Double simulation.





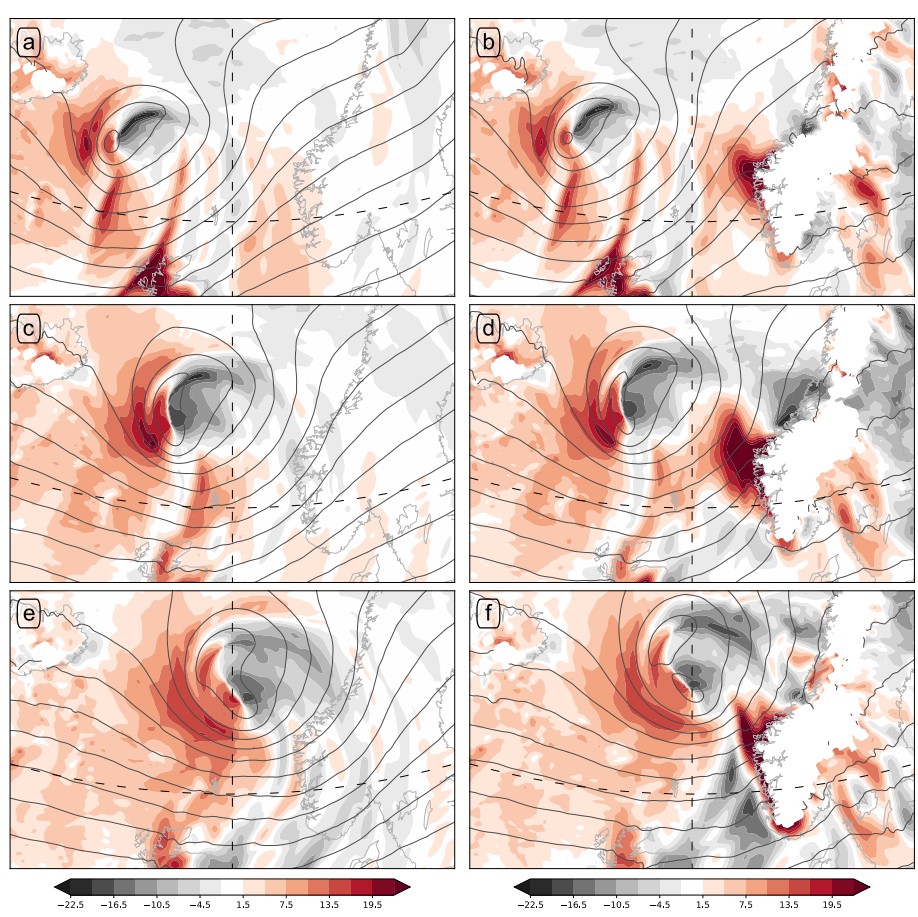

**Figure 8.** Mass transport [$\mathrm{kg\,s^{-1}\,m^{-2}}$] perpendicular to geopotential isolines at 850 hPa, with positive values indicating transport towards lower geopotential. The rows show the time evolution for (a,b) 00 UTC, (c,d) 03 UTC and (e,f) 06 UTC. The left column is based on a WRF simulation in which the Scandinavian orography as been removed and converted to ocean, the right column on a WRF simulation in which the Scandinavian orography is twice its original height.

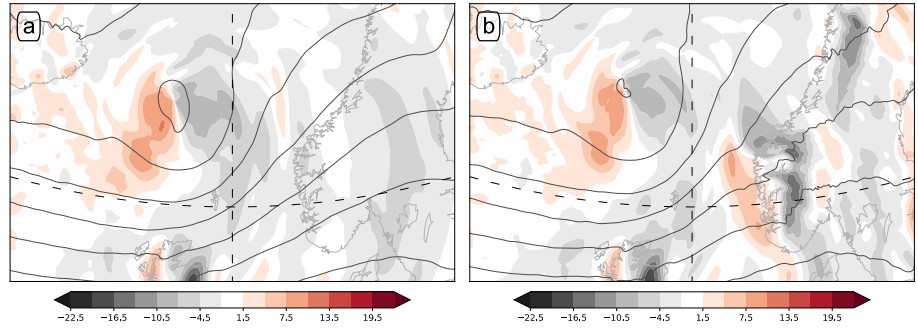

**Figure 9.** As Fig. 8c,d, but for mass transport at 500 hPa.

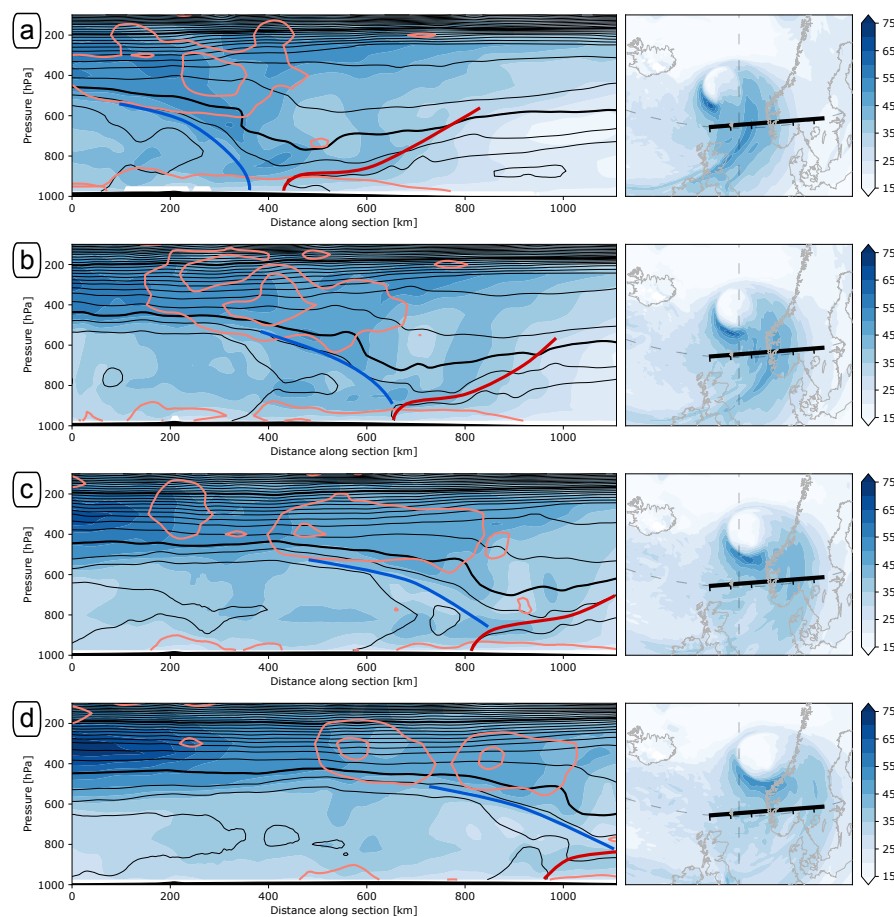

**Figure 10.** Cross section of wind speed (shading, $\mathrm{m\,s^{-1}}$), equivalent potential temperature (black contours with interval 5 K, 300 K and 350 K thickened) and ageostrophic horizontal wind with contours at 15, 25 and 35 $\mathrm{m\,s^{-1}}$. The blue and red lines mark subjectively analysed cold and warm fronts, respectively. The maps in the right column show wind speed at 850 hPa as well as the location of the cross section with ticks every 200 km. The rows show (a) 03 UTC, (b) 05 UTC, (c) 07 UTC, and (d) 09 UTC on 1 January 1992 for the Ocean simulation.



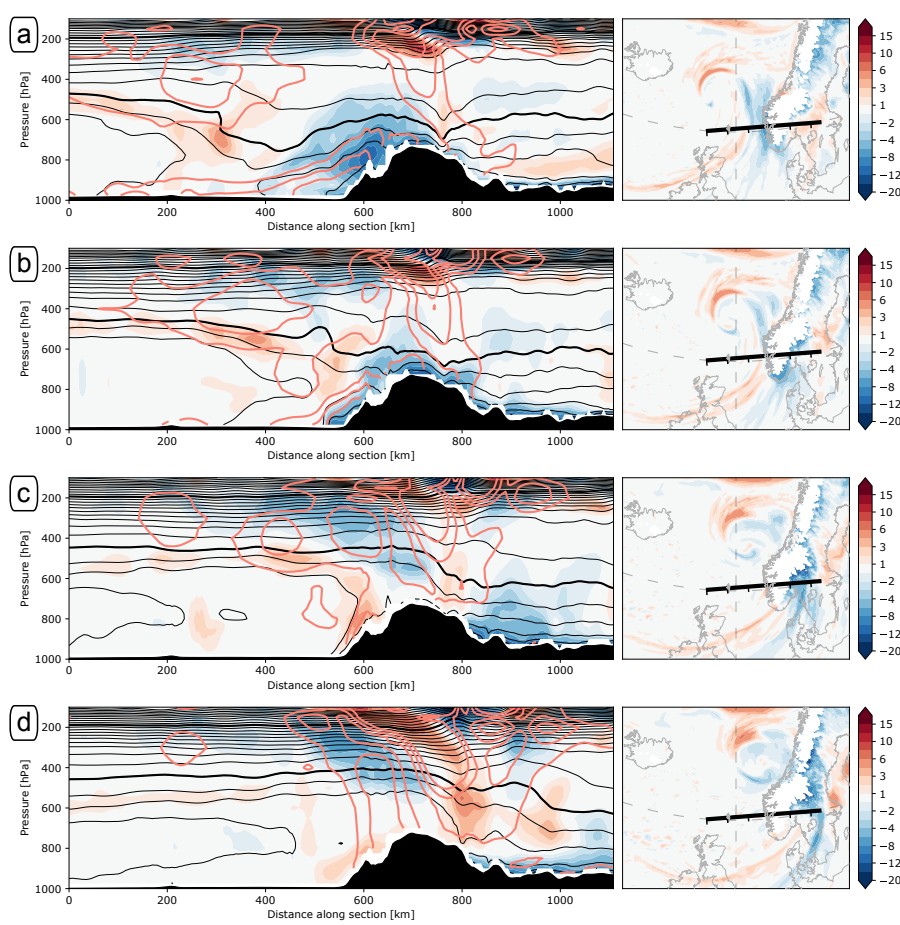

**Figure 11.** As Fig. 10, but showing in shading the equivalent potential temperature difference in Kelvin between the Double and the Ocean simulation. Contours show equivalent potential temperature and ageostrophic wind speed as in Fig. 10, but for the Double simulation.

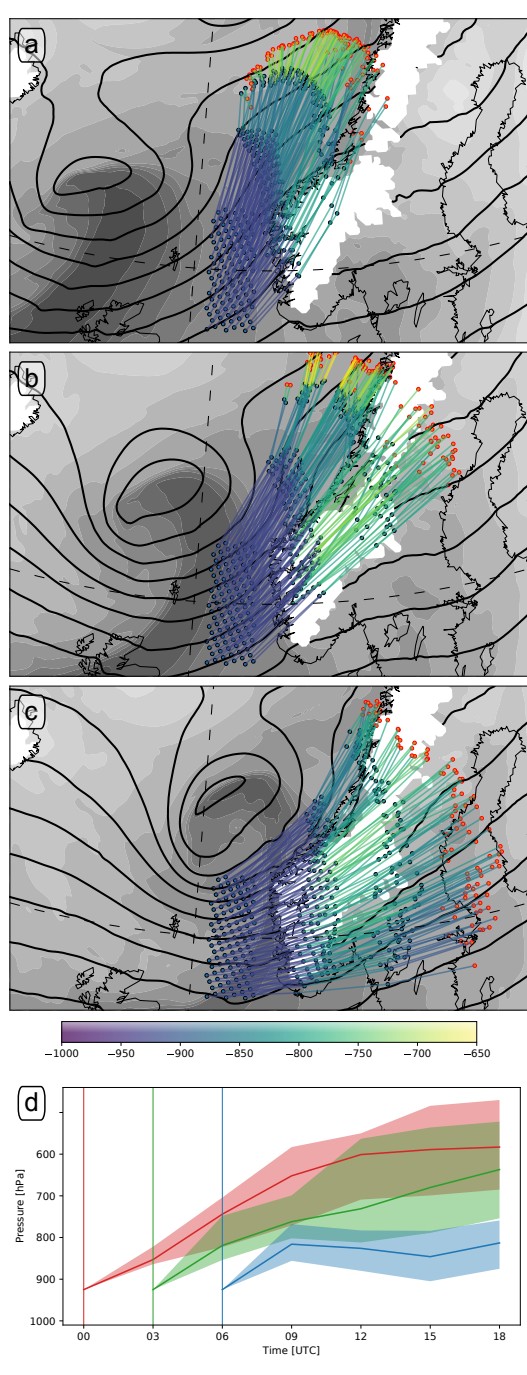

**Figure 12.** Forward trajectories released from the box 58.0-62.0°N, 1.0-4.0°E at 925 hPa at (a) 00 UTC, (b) 03 UTC and (c) 06 UTC. The trajectory segments each cover 3 hours, and they are coloured by pressure [hPa] of the *preceding* time step. The grey shading and black contours shows $\theta_e$ and geopotential, respectively at 925 hPa, at the release time of the trajectories with the same contour interval and limits as in Fig. 2. The bottom panel (d) shows the median pressure evolution of the above parcels. The respective transparent shading indicates variability between the trajectories by the 15 and 85 percentiles.