# Peer review of "Front-orography interactions during landfall of the New Year's Day Storm 1992"

_Weather and Climate Dynamics, 2019_

## Referee Comment (RC1) · Anonymous Referee #1 · 11 Dec 2019

This paper re-examines the notorious New Year's Day storm that struck Western Norway on January 1 1992, and led Grønås (1995) to introduce the 'poisonous sting at the end of the tail' to the meteorological literature, now referred to as a sting jet. The main focus is on the effect the Scandinavian mountains had on the thermodynamic structure of the fronts in this storm, although the paper also examines whether the mountains also affected the development of the strong winds that struck the Norwegian coast (the answer to this is a resounding no – the same winds would have developed regardless of the mountains). The paper is an interesting study but needs considerable polishing to be a useful addition to the meteorological literature.

[Figure]

My general criticism of the paper is that the concluding section does not link back to the introduction, and to the existing literature on the effect of orography on fronts. As a result it is impossible to see what is new in this paper and what corroborates (or contradicts) previous results (other than the null effect on the CCB wind maximum). The summary and conclusions need re-writing to place the current results in context, and should concentrate on verifiable results rather than speculation (e.g. l.319-20, 328-32). This is a well-established area of research in meteorology. Only if the authors can show a genuine novel result should this paper be published.

Specific comments 1. Section 3. The figures that accompany this section show fields from both NORA and WRF, but the text does not make it clear which model field is being discussed. I would have thought that the reanalysis would be closer to reality than a free-running model so the synoptic discussion should be confined to NORA, making it a little easier to follow. I'm not sure why you need all the WRF graphs as their only purpose as far as I can tell is to satisfy the reader that the WRF simulation looks sufficiently similar to the reanalysis. Section 3.4 is far too superficial to require 13 figure panels (figs 2, 3, 4).

2. Line 256-260. The authors propose that IGWs are responsible for the effect of orography on mass transport at 500 mb. This need not be so: the mountains change the thermodynamic fields at the lower levels and therefore the height field at 500 mb. Mountain waves can only impart momentum to the flow if they break, and as they are fixed relative to the topography their effect would be to slow the winds at 500 mb. That would disturb the geostrophic balance, suggesting a flow towards low pressure, which is the opposite of that shown in fig 9b. In the absence of any evidence this paragraph is pure speculation, quite probably wrong, and should be removed.

3. The same unwarranted speculation continues in the first paragraph of the next section, which should either be removed or solid evidence be presented for this conjecture.

Typos

Line 71 resolution of 10 km

Line 180 'warm-air seclusion suggests that . . . cyclone core may be dynamically linked' – correlation does not prove causation

Line 225 fronts fronts

Line 265 vertical wind no longer shows

---

## Referee Comment (RC2) · Anonymous Referee #2 · 16 Dec 2019

General comments:

The interactions of the fronts of the famous "New year's day storm" from 1992 with the Norwegian orography is investigated in this paper through analysis of the NORA10 reanalysis data and three simulations with the WRF model (a control and two simulations with modified orography). The study is presented well with detailed analysis and should be of interest to readers of this journal. It would be strengthened by the results being placed more firmly in the context of those from other studies. I recommend that the authors consider my, mainly minor, comments below.

[Figure]

Major specific comments:

**Summary and conclusions:** This section is rather brief and just summarises the results from the paper rather than placing these results in the context of other studies. Please link the results to those from the other studies discussed in the introduction.

Minor specific comments:

**Section 2** You say how many vertical model levels there are in the NORA10 hindcast and WRF simulations but please add information about the model top and the midtropospheric vertical model level spacing (i.e. before interpolating to pressure levels).

**L69:** I got confused by these two timescales. Initially I thought that $UTC_{Sat}$ was the actual time whereas $UTC_{No}$ was an adjusted time to take account of the 1.5 hr time displacement in the hindcast such that e.g. 0000 $UTC_{No}$ would actually be 0130 UTC in the run. From the caption of Fig 5 I worked out that both times are the "actual" times, but that $UTC_{No}$ indicates the time in the lagging hindcast run. It might be easier to remove this notation but instead just note the times corresponding to the same stage of the evolution in the hindcast run where required. Relating to this point, how does the timing of the evolution in the WRF control simulation match to the satellite inferred development, is it better than that in the NORA10 hindcast? If so, do you have any idea why? The warm air seclusion is not as warm in the WRF control run as in the NORA10 hindcast, is this important?

**L127:** Presumably the main reason that the absolute wind is weaker along the developing bent-back front than along the cold front is because it is in the opposite direction to the motion of the cyclone i.e. although the earth-relative winds are weak, the cyclone-relative winds would be stronger.
**Fig 5:** What is the meaning of the blue spots in panels (a) and (c)?

**L187:** Do you mean to refer to Fig. e,f rather than d,e? The low-level jet associated with the bent-back front is a long way from the coast in panel d.

**L196:** It would be helpful to mark the regions of descent (Fig 4f) and patchy up and down drafts (fig 4e) in the corresponding figs. Note that difference in the vertical motion dynamics referred to in the text is only really clear in panels e,f and not also at the later time shown in panels g,h.

**L207:** Move shorthand terminology for the runs to section 2.1 as the terminology is first used there (L89: "in the Ocean simulation").

**L211:** Which bit of the warm sector are you referring to when you mention the "on going occlusion process"? Is it the weak filament of warm air over Scandinavia in Fig 7a which isn't there in Fig 2h, so indicating a slower occlusion process in the Ocean simulation?

**L215:** You say that the detected bent-back front extends further around the warm air seclusion in the Ocean simulation. Which run are you comparing with here though, the control run or that with Double orography? The extent of the bent-back front on the northern side of the seclusion is greatest in the control run (Fig 2h).

**Fig 7 caption** I think you mean "as Fig 2h" as both panels show results from the WRF runs (whereas Fig 2g is for the NORA10 hindcast).

**L244:** Can you provide some extra information to help me locate where you mean by "the cold sector"?

**L258:** Given that air is pretty much incompressible is it really surprising that the ascent and mass transport that occurs as the flow impinges on the orography is pretty much instantaneously seen at all levels (including the weak ascent at 500 hPa)?

**Fig 11:** Please clarify in the text whether the maps in the right hand column are $\theta_e$ differences (as they appear to be from comparison with the left hand column plots) or wind speed (as in Fig 10 which Fig 11 is supposed to mirror in structure).

**L291:** Please add some detail about how the trajectories were calculated. From Fig 12d I think the trajectories all run to 18 UTC and are started at 0, 3 and 6 UTC. But each panel seems to have 4 sets of circle markings along the trajectories— what times are these at? The caption refers to 3 hour trajectory segments so does e.g., Fig 12a just show the trajectories from 0 to 9 UTC? The caption states that the trajectories are coloured by the pressure of the preceding timestep— does this mean that the colour of the first segment of the trajectories in panel a relates to the pressure of these trajectories at 0 UTC? Why are the pressures then all negative in the colour bar under panel c? Finally, why are the last set of circle markers along the trajectories red whereas the other ones seem to match the colour of the trajectory segments?

Technical errors:

**L23:** Change "have" to "have had".

**L27:** Change "by to the" to "by the".

**L71:** Add "of" before "10 km".

**L137:** Change "long" to "along".

**L170:** Change "front" to "fronts".

**L197:** Change "well compared" to "compared well".

**L202:** Change to either "a clear orographic influence" or "clear orographic influences".

**L224:** Add "the" before "somewhat".

**L233:** I think you mean "sector" rather than "section" here.

**L300 and L301:** Change "begin" to "beginning".

---

## Author Comment (AC1) · 26 Jan 2020

**Response to referees – "Front-orography interactions during landfall of the New Year'ss Day Storm 1992"**

**C. Spensberger and S. Schemm**

**26 January 2020**

We thank both referees for their constructive and detailed criticism. We agree with most of the comments and will adapt the manuscript to mitigate concerns and correct errors. Specific replies to the issues raised are below. Our replies are marked blue.

**Referee 1**

**General criticism** My general criticism of the paper is that the concluding section does not link back to the introduction, and to the existing literature on the effect of orography on fronts. As a result it is impossible to see what is new in this paper and what corroborates (or contradicts) previous results (other than the null effect on the CCB wind maximum). The summary and conclusions need re-writing to place the current results in context, and should concentrate on verifiable results rather than speculation (e.g. l.319-20, 328-32). This is a well-established area of research in meteorology. Only if the authors can show a genuine novel result should this paper be published.

We will rework the introduction and conclusions to better link these sections and make the novel aspects of our study more visible. To this end, we will make reference in the concluding section to three specific research questions posed in the reworked introduction:

1. Did the landfall of the storm affect the formation of its poisonous tail?

2. Did the landfall accelerate the formation of a warm-air seclusion?

3. Did the landfall affect the track of the New Year's Day Storm?

4. How did the landfall affect the air streams and conveyor belts around the New Year's Day Storm?

Further, the present case study adds to the literature in two ways:

1. Our results indicate that the cyclone core and its warm sector might evolve into dynamically independent entities with the onset of the occlusion process. This presents a potential paradigm shift, as most of the literature we are aware of discuss and thus implicit consider occluded cyclones as one system.

2. Our results indicate that IGW activity might have played a role in eroding the fronts and communicating the orographic impact to the middle and higher troposphere. In our view that is an interesting novel perspective, as the role of IGWs in front-orography interactions has so far not been systematically assessed.

**Section 3** The figures that accompany this section show fields from both NORA and WRF, but the text does not make it clear which model field is being discussed. I would have thought that the reanalysis would be closer to reality than a free-running model so the synoptic discussion should be confined to NORA, making it a little easier to follow. I'm not sure why you need all the WRF graphs as their only purpose as far as I can tell is to satisfy the reader that the WRF simulation looks sufficiently similar to the reanalysis. Section 3.4 is far too superficial to require 13 figure panels (figs 2, 3, 4).

The NORA10 data set is a set of short hindcasts without data assimilation. Technically, NORA10 is closer related to our WRF simulation than it is to a reanalysis product. More specifically, NORA10 is a blend of members within a sequence of short (<9h lead time) mesoscale model integrations initialised

and forced by ERA-40. Based on the available data and observations, it is difficult to identify which one of the two data sets is closer to reality, as both have strengths and weaknesses. We therefore prefer to keep both datasets for the discussion of the horizontal wind field (Fig. 2 and Fig. 3). But, as is suggested by the reviewer, we confine the discussion of the thermal structure and pressure tendency (Fig. 4 and Fig. 6) to NORA10 data.

**Line 256-260/Unwarranted speculation** The authors propose that IGWs are responsible for the effect of orography on mass transport at 500 mb. This need not be so: the mountains change the thermodynamic fields at the lower levels and therefore the height field at 500 mb. Mountain waves can only impart momentum to the flow if they break, and as they are fixed relative to the topography their effect would be to slow the winds at 500 mb. That would disturb the geostrophic balance, suggesting a flow towards low pressure, which is the opposite of that shown in fig 9b. In the absence of any evidence this paragraph is pure speculation, quite probably wrong, and should be removed.

The same unwarranted speculation continues in the first paragraph of the next section, which should either be removed or solid evidence be presented for this conjecture.

We thank the reviewer for sharing these concerns. According to the literature, IGWs can impart momentum to the flow also when they non-linearly interact with each other or the background state on which they propagate (e.g. Einaudi et al., 1978; Plougonven and Zhang, 2014; Fritts et al., 2016). We further agree with the physical reasoning of the reviewer that the waves would slow down flow at 500 hPa, and thus induce flow towards the cyclone center. That is also exactly what we see along the Norwegian west coast in Fig 9b, positive values indicate flow towards lower geopotential (cf. caption of Fig. 8).

Finally, it is true that the changing thermal structure will change the geopotential structure aloft. The winds will thereafter however only change through geostrophic adjustment, which is a comparatively slow process (e.g. Blumen, 1972).

In response to the concerns raised by both reviewers, we will rework the corresponding section to include additional arguments as outlined above. The wording will be toned down and we highlight that we present a hypothesis. We remove speculations from the abstract and concluding section.

**Typos**

Thanks for pointing these out, we will correct.

**Referee 2**

**General comments** The interactions of the fronts of the famous New years day storm from 1992 with the Norwegian orography is investigated in this paper through analysis of the NORA10 reanalysis data and three simulations with the WRF model (a control and two simulations with modified orography). The study is presented well with detailed analysis and should be of interest to readers of this journal. It would be strengthened by the results being placed more firmly in the context of those from other studies. I recommend that the authors consider my, mainly minor, comments below.

**Summary and conclusions** This section is rather brief and just summarises the results from the paper rather than placing these results in the context of other studies. Please link the results to those from the other studies discussed in the introduction.

We agree with both referees in that the introduction, results and conclusions sections need to be better linked. We will rework these sections and introduce specific research questions in the introduction to which we explicitly refer in the discussion of the results and the concluding section.

We will further link our findings better to the results of earlier studies, for example:

- Braun et al. (1997), Kljun et al. (2001), and Neiman et al. (2004) as contrasting case studies for cold front-orography interactions.

- Doyle and Bond (2001) as a contrasting case study of warm front-orography interactions.

- Davies (1984), Blumen and Gross (1987), and Egger (1992) as conceptual context for front-orography interactions in general.

- Einaudi et al. (1978), Plougonven and Zhang (2014), and Fritts et al. (2016) as context for the potential role of IGWs in our case study.

Most of these citations are/will be put into context in the reworked introduction, thus providing further links between the introduction, discussion and conclusion sections.

**Section 2** You say how many vertical model levels there are in the NORA10 hindcast and WRF simulations but please add information about the model top and the mid-tropospheric vertical model level spacing (i.e. before interpolating to pressure levels).

Both models have the hybrid model levels with the top model level at 10 hPa, and about 35-40 hPa resolution in the middle troposphere (at a surface pressure of about 1000 hPa). We will add this information to the descriptions of the model setup.

**L69** I got confused by these two timescales. Initially I thought that $UTC_{Sat}$ was the actual time whereas $UTC_{No}$ was an adjusted time to take account of the 1.5 hr time displacement in the hindcast such that e.g. 0000 $UTC_{No}$No would actually be 0130 UTC in the run. From the caption of Fig. 5 I worked out that both times are the actual times, but that $UTC_{No}$ indicates the time in the lagging hindcast run. It might be easier to remove this notation but instead just note the times corresponding to the same stage of the evolution in the hindcast run where required. Relating to this point, how does the timing of the evolution in the WRF control simulation match to the satellite inferred development, is it better than that in the NORA10 hindcast? If so, do you have any idea why? The warm air seclusion is not as warm in the WRF control run as in the NORA10 hindcast, is this important?

We agree with the referee that the two different time axes can be confusing. With this comment in mind, we have come to the conclusion that it will be clearer for the reader if we only mention the differences in timing when discussing Fig. 2 rather than introducing different time axes.

The WRF simulation seems to lag a bit less behind the satellite imagery than the NORA10 data. Testing different initialisation times for the WRF simulation, we noted that the simulated intensity of the storm is quite sensitive to the initialisation time. It seems plausible that the sensitivity in intensity goes hand in hand with a sensitivity in timing. Further, the WRF version we used for our simulations is considerably newer than the version of HIRLAM used to create the NORA10 hindcasts. Amongst other parameterisation, in particular different representations of moist diabatic processes could lead to a more-or-less explosive development of the storm and thus timing differences.

We think the slightly warmer warm seclusion in NORA10 has no substantial influence on the front-orography interactions around the storm's warm sector, because the simulations compare well for other parts of the storm. The simulated wind speeds along the coast are in reasonable agreement with observations for both simulations. They both have their strength and weaknesses and it is thus important to have them both in one manuscript.

**L127** Presumably the main reason that the absolute wind is weaker along the developing bent-back front than along the cold front is because it is in the opposite direction to the motion of the cyclone i.e. although the earth-relative winds are weak, the cyclone-relative winds would be stronger.

We agree with that comment and will add it to the discussion in the manuscript.

**Fig 5** What is the meaning of the blue spots in panels (a) and (c)?

Thanks for pointing those out. Those were the locations of synthetic soundings we included in an earlier revision of the manuscript. We will remove them.

**L187** Do you mean to refer to Fig. e,f rather than d,e? The low-level jet associated with the bent-back front is a long way from the coast in panel d.

Thanks for pointing out this odd reference. It is actually supposed to refer to Fig. 3g,h.

**L196** It would be helpful to mark the regions of descent (Fig 4f) and patchy up and down drafts (fig 4e) in the corresponding figs. Note that difference in the vertical motion dynamics referred to in the text is only really clear in panels e,f and not also at the later time shown in panels g,h.

We agree with the referee and mark the regions in Fig 3e,f and refer to these panels instead.

**L207** Move shorthand terminology for the runs to section 2.1 as the terminology is first used there (L89: in the Ocean simulation).

Thanks for pointing this out, we will adapt.

**L211** Which bit of the warm sector are you referring to when you mention the on going occlusion process? Is it the weak filament of warm air over Scandinavia in Fig 7a which isnt there in Fig 2h, so indicating a slower occlusion process in the Ocean simulation?

We refer to the rather long and thin warm sector between about 60N and the Norwegian coast line near the warm seclusion. We will make this reference more specific in the text.

**L215** You say that the detected bent-back front extends further around the warm air seclusion in the Ocean simulation. Which run are you comparing with here though, the control run or that with Double orography? The extent of the bent-back front on the northern side of the seclusion is greatest in the control run (Fig 2h).

Thanks for pointing out this inconsistency! We will either remove this remark entirely or add the comparison to the control simulation.

**Fig 7 caption** I think you mean as Fig 2h as both panels show results from the WRF runs (whereas Fig 2g is for the NORA10 hindcast).

We agree with the referee and will correct.

**L244** Can you provide some extra information to help me locate where you mean by the cold sector?

We refer to narrow north-south oriented feature just south of the cyclone core in Figs 8a-d. We will mark it on the maps and refer to this mark.

**L258** Given that air is pretty much incompressible is it really surprising that the ascent and mass transport that occurs as the flow impinges on the orography is pretty much instantaneously seen at all levels (including the weak ascent at 500 hPa)?

It seems very plausible that the forced ascent in the lowest troposphere is at least in part the dynamical origin of the signal we observe at the 500 hPa-level. However, our trajectory analysis suggest that the first parcels of near-surface air are only reaching the 500 hPa level after 9-12 hours of ascent, such that vertical mass transport is too slow to account for the vertical communication of the anomaly signal. The dipole in the mass flux appears nearly instantaneously, and extends throughout most of the troposphere. In the light of the strong wave signal seen in the vertical velocity field, it seems plausible to us that IGWs played a role in propagating the signal upwards from the lowest troposphere.
    We will rework this section and include the arguments offered here and in response to referee 1 in the manuscript. We will clearly mark the section as a hypothesis and reduce its prominence in the abstract and conclusions.

**Fig 11** Please clarify in the text whether the maps in the right hand column are $\theta_e$ differences (as they appear to be from comparison with the left hand column plots) or wind speed (as in Fig 10 which Fig 11 is supposed to mirror in structure).

These are $\theta_e$ differences. We will clarify the caption to avoid this ambiguity.

**L291** Please add some detail about how the trajectories were calculated. From Fig. 12d I think the trajectories all run to 18 UTC and are started at 0, 3 and 6 UTC. But each panel seems to have 4 sets of circle markings along the trajectories what times are these at? The caption refers to 3 hour trajectory segments so does e.g., Fig. 12a just show the trajectories from 0 to 9 UTC? The caption states that the trajectories are coloured by the pressure of the preceding timestep does this mean that the colour of the first segment of the trajectories in panel a relates to the pressure of these trajectories at 0 UTC? Why are the pressures then all negative in the colour bar under panel c? Finally, why are the last set of circle markers along the trajectories red whereas the other ones seem to match the colour of the trajectory segments?

Thanks for pointing out this inconsistency between the trajectory map and height plot. The circles mark the trajectory locations at 3-hourly intervals. So you have been interpreting the Figure correctly, for panel (a) the circles mark the locations at 00, 03, 06 and 09 UTC.
    You're interpreting the color of the trajectory segments correctly. The negative pressures are a typo, resulting from a struggle with the plotting library. The absolute numbers give pressure in hPa. Thanks for pointing this out!
    The final circles are red rather than black just for visual distinction from the previous time step. Some trajectories (for example those close to the cyclone core) do not move much in the last of the 3-hour intervals, such that the circles would become ambiguous were they black, too. We will add small circles to the trajectory height plot to make that correspondence between the panels more intuitive. We will also add the missing information about the circles to the caption.

**Technical errors**

Thanks for pointing these out, we will correct.

**References**

Blumen, W.: Geostrophic adjustment, Reviews of Geophysics, 10, 485–528, https://doi.org/10.1029/RG010i002p00485, 1972.

Einaudi, F., Lalas, D. P., and Perona, G. E.: The role of gravity waves in tropospheric processes, pure and applied geophysics, 117, 627–663, https://doi.org/10.1007/BF00879972, 1978.

Fritts, D. C., Smith, R. B., Taylor, M. J., Doyle, J. D., Eckermann, S. D., Drnbrack, A., Rapp, M., Williams, B. P., Pautet, P.-D., Bossert, K., Criddle, N. R., Reynolds, C. A., Reinecke, P. A., Uddstrom, M., Revell, M. J., Turner, R., Kaifler, B., Wagner, J. S., Mixa, T., Kruse, C. G., Nugent, A. D., Watson, C. D., Gisinger, S., Smith, S. M., Lieberman, R. S., Laughman, B., Moore, J. J., Brown, W. O., Haggerty, J. A., Rockwell, A., Stossmeister, G. J., Williams, S. F., Hernandez, G., Murphy, D. J., Klekociuk, A. R., Reid, I. M., and Ma, J.: The Deep Propagating Gravity Wave Experiment (DEEPWAVE): An Airborne and Ground-Based Exploration of Gravity Wave Propagation and Effects from Their Sources throughout the Lower and Middle Atmosphere, Bulletin of the American Meteorological Society, 97, 425–453, https://doi.org/10.1175/bams-d-14-00269.1, 2016.

Plougonven, R. and Zhang, F.: Internal gravity waves from atmospheric jets and fronts, Reviews of Geophysics, 52, 33–76, https://doi.org/10.1002/2012RG000419, 2014.

---

## Author Response (AR1)

**Response to referees – "Front-orography interactions during landfall of the New Year's Day Storm 1992"**

**C. Spensberger and S. Schemm**

**26 January 2020**

We thank both referees for their constructive and detailed criticism. We agree with most of the comments and adapted the manuscript to mitigate concerns and correct errors. Specific replies to the issues raised are below. Our replies are marked blue.

**Referee 1**

**General criticism** My general criticism of the paper is that the concluding section does not link back to the introduction, and to the existing literature on the effect of orography on fronts. As a result it is impossible to see what is new in this paper and what corroborates (or contradicts) previous results (other than the null effect on the CCB wind maximum). The summary and conclusions need re-writing to place the current results in context, and should concentrate on verifiable results rather than speculation (e.g. 1.319-20, 328-32). This is a well-established area of research in meteorology. Only if the authors can show a genuine novel result should this paper be published.

We revised the introduction and conclusions to better link these sections and make the novel aspects of our study more visible. To this end, we make reference in the concluding section to three specific research questions posed in the revised introduction:

- 1. Did the landfall of the storm affect the formation of its poisonous tail?
- 2. Did the landfall affect the track of the New Year's Day Storm?
- 3. How did the landfall affect the warm and cold fronts of the New Year's Day Storm?

These questions reflect the discussions in sections 4.1-4.3 and are taken up again in the summary-andconclusions section.

Further, the present case study adds to the existing literature in two ways:

- 1. Our results indicate that the cyclone core and its warm sector might evolve into dynamically independent entities with the onset of the occlusion process. This presents in our view a genuinely novel result, as most of the literature we are aware of discuss and thus implicit consider occluded cyclones as one system.
- 2. Our results suggest that IGW activity might have played a role in eroding the fronts and communicating the orographic impact to the middle and higher troposphere. In our view that is an interesting novel perspective, as the role of IGWs in front-orography interactions has so far not been systematically assessed.

Section 3 The figures that accompany this section show fields from both NORA and WRF, but the text does not make it clear which model field is being discussed. I would have thought that the reanalysis would be closer to reality than a free-running model so the synoptic discussion should be confined to NORA, making it a little easier to follow. I'm not sure why you need all the WRF graphs as their only purpose as far as I can tell is to satisfy the reader that the WRF simulation looks sufficiently similar to the reanalysis. Section 3.4 is far too superficial to require 13 figure panels (figs 2, 3, 4).

The NORA10 data set is a set of short hindcasts without data assimilation. Technically, NORA10 is closer related to our WRF simulation than it is to a reanalysis product. More specifically, NORA10 is a blend of members within a sequence of short (

[revised manuscript text omitted]

---

## Referee Report (RR1)

Comments on revised manuscript by Spensberger and Schemm,

Front-orography interactions during landfall of the New Year's Day storm 1992

This paper is much improved from the previous version. In particular the introduction and conclusions are now properly connected and it is clearer what the paper has to say that is new.

I am however still concerned about the argument regarding gravity waves. It seems to hinge on a strip of pink colour upstream of the Norwegian coast in fig 9b, which is the 500 mb field of mass transport perpendicular to isolines at 03 UTC, from the doubled-orography WRF run. The pink strip denotes flow towards lower geopotential. The vertical velocity field from this model run isn't shown, but assuming it resembles that with the original orography, we should compare 9b to 4d. There are clearly no gravity waves in the location of the pink strip in fig 4d, and where gravity waves are found (over the whole Norwegian land mass, not just the Scandes mountains) the mass flux in 9b is away from the cyclone centre. This not what I would expect if the waves were breaking and decelerating the flow. The paper waves this aside with a sentence I do not understand: 'A IGW-induced deceleration would in turn give rise to a geostrophic adjustment process with an initial acceleration towards the cyclone core, consistent with Figure 9'.

The mass flux structure in 9b does indeed suggest a local trough due to ageostrophic flow in this region, roughly on the scale of the Scandes, reminiscent of lee cyclogenesis, though it's hard to see that in the geopotential. In turn that suggests that the response of the atmosphere to the perturbation of the mountain range is, first and foremost, to set up a deformation on the scale of the mountain range that extends into the upper troposphere. This indeed appears 'simultaneously' throughout the troposphere because the mechanism is gravity wave propagation, not geostrophic adjustment. But the crucial point is that this has nothing to do with the much smaller-scale orographic waves shown in fig.4: it is the response of the fluid field as a whole to the orography.

The hypothesis that the smaller-scale waves transferred momentum to the flow requires that they break – i.e. reduce in amplitude with height or encounter a critical level. The authors could easily have checked that by examining how the wave field varies with height, but they have not done so, preferring instead to speculate.

Section 4.4 starts by discounting the possibility that the response at 500 mb was due to advection, concluding that it can't be. This is complete red herring – of course it's nothing to do with the physical movement of air parcels as the orographic perturbations move (rapidly) through the fluid. Geostrophic adjustment is a process whereby the atmosphere adjusts its wind and pressure fields towards geostrophy when it is forced away from dynamic balance. Again this is irrelevant to a process which is being forced by the orography, which is maintaining the dynamic imbalance.

There are copious books and papers dealing with flow over orography which the authors should consult. There they will see how the flow patterns adjust to the presence of the orography, extending up through the atmosphere 'almost-instantaneously'.

I found a few typos:
l.2 Coast
l. 153 Wind field is shown in Fig 3 not 2
l. 165 frontal structure
l.191 The blue circles in fig 4e,f are over the ocean, not over the lee of the Scandes
l.220 'is there' rather than 'there is' (an English subtlety which I can't explain!)
l.334 accelerated
l.353 too

---

## Referee Report (RR2)

**2nd Review of 'Front-orography interactions during landfall of the New Year's Day storm 1992' by Spensberger and Schemm**

**General comments:**

The interactions of the fronts of the famous "New year's day storm" from 1992 with the Norwegian orography is investigated in this paper through analysis of the NORA10 reanalysis data and three simulations with the WRF model (a control and two simulations with modified orography). The study is presented well with detailed analysis and should be of interest to readers of this journal. The authors have addressed my previous concerns about the paper and I have just a few minor additional comments.

**Minor specific comments:**

**L153** Here you refer to Fig 2e,f as evidence that the "southwesterlies in the warm sector at 850 hPa are not evident in the wind field in the lee of the Scandes". However, this figure doesn't show wind field but instead shows $\theta_e$ and geopotential height. Are you inferring the (geostrophic) wind field from the geopotential height?

**L215** Here you say that the $\theta_e$ gradient is less locally confined on the northern side of the seclusion *without* orographic influence. However, as you state earlier in the same sentence (and as shown in Fig 7), it is in the double orography simulation that the "bent-back front extends less far around the warm-air seclusion" (compared to the control and no orography (control) simulation). Hence you seem to be contradicting yourselves.

**L230, 231** Here you refer to the "incipient" cyclone (in 2 places). This word doesn't really make sense though given that I think you're referring to the existing mature cyclone and incipient means beginning to happen/develop. Do you mean "existing" cyclone? Also in L282, do you really mean "incipient"?

**Technical errors:**

**Abstract, L2** "Cost" should be "coast"

**L216** "are" should be "is".

**L353** "emerges to rapidly" should be "emerges too rapidly".

---

## Author Response (AR2)

**2nd response to referees – "Front-orography interactions during landfall of the New Year's Day Storm 1992"**

**C. Spensberger and S. Schemm**

**26 March 2020**

We thank both referees for considering the manuscript a second time. We are happy to see that the referees consider the manuscript to be much improved. In the light of the renewed criticism of referee 1 of our discussion of the role of IGWs, we decided to remove this section from the present manuscript. We will instead pursue these ideas in a dedicated study.

We further thank pointing out some remaining typos and technical errors, we corrected them. Responses to the specific comments of reviewer 2 are given below in blue.

**Specific comments of referee 2**

L153 Here you refer to Fig 2e, f as evidence that the "southwesterlies in the warm sector at 850 hPa are not evident in the wind field in the lee of the Scandes". However, this figure doesn't show wind field but instead shows  $\theta_e$  and geopotential height. Are you inferring the (geostrophic) wind field from the geopotential height?

Thanks for pointing out this erroneous reference. We meant to refer to Figure 3c and adapted accordingly.

**L215** Here you say that the  $\theta_e$  gradient is less locally confined on the northern side of the seclusion without orographic influence. However, as you state earlier in the same sentence (and as shown in Fig 7), it is in the double orography simulation that the "bent-back front extends less far around the warmair seclusion" (compared to the control and no orography (control) simulation). Hence you seem to be contradicting yourselves.

Thanks for pointing out this mistake, we did not intend the double-negation. The *without* should have been a *due to*. We corrected.

L230, 231 Here you refer to the "incipient" cyclone (in 2 places). This word doesn't really make sense though given that I think you're referring to the existing mature cyclone and incipient means beginning to happen/develop. Do you mean "existing" cyclone? Also in L282, do you really mean "incipient"?

Thanks for pointing out this erroneous word use. With this word, we intended to express the motherdaughter kind of relation between the "incipient" and secondary cyclone. We now replaced "incipient" with "pre-existing".

[revised manuscript text omitted]
. We will examine the potential role of these waves for both the rapid decay of the warm and cold fronts over the Scandes - The cross-frontal length seale is comparable to the wave length, such that interactions are plausible. To underpin these more tentative results, and generalise them beyond the case study presented here, we require a more systematic assessment of the
 360 role of IGWs for front-orography interactions and the emergence of the anomalous mass flux along the Norwegian coast line in

a follow-up study.

[revised manuscript text omitted]